# When the Land Sings: Reconstructing Prehistoric Environments Using Evidence from Quaternary Geology and Geomorphology, with Examples Drawn from Fluvial Environments in the Nile and Son Valleys

Martin Williams 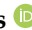

Department of Earth Sciences, University of Adelaide, Adelaide 5005, Australia; martin.williams@adelaide.edu.au

**Abstract:** Geomorphic evidence from rivers and lakes can help explain past changes in the locations of archaeological sites as well as environmental and climatic changes in their catchment areas. Examples drawn from the Blue and White Nile valleys in northeast Africa and from the Son and Belan valleys in north-central India reveal how Quaternary climatic fluctuations in the headwaters of these rivers are reflected in changes in river channel patterns downstream as well as in the type of sediment transported. Soils and sediments that contain prehistoric and historic artefacts can be analysed to show the type of environment in which the artefacts accumulated. Beds of volcanic ash may preserve former landscapes and their fossil remains and can provide a synchronous time marker against which to assess changes in the archaeological record. The pattern and tempo of past sea level fluctuations has controlled the distribution of coastal archaeological sites and helps to explain the absence of certain Holocene Neolithic sites in southeast Asia. Disturbance of archaeological sites by plants and animals, especially termites in tropical regions, can affect the stratigraphic and chronological integrity of the site.

**Keywords:** dunes; desert dust; fossil soils; geoarchaeology; geomorphology; lakes; loess; prehistoric environments; Quaternary; rivers; sediments; soils



## 1. Introduction

Geoarchaeology is the application of insights derived from the earth sciences to resolving archaeological problems [1–6]. Such insights may derive from biological evidence, such as fossil remains and genetic evidence, physical evidence such as rocks, sediments, soils and landforms, chemical evidence such as the isotopic composition of a sample and chronological evidence, including radiocarbon and luminescence dating [7–11]. The wider the array of independent techniques that are used when seeking to interpret archaeological evidence in its environmental context, the more solidly based will be the final conclusions. This contribution looks at one type of evidence, namely, the evidence derived from geomorphology. For the purposes of this review, we can define geomorphology very simply as the study of landforms and the processes responsible for their formation [11–13]. These processes include weathering, erosion and sediment deposition by water, wind and ice. Geomorphological evidence includes various types of landforms, such as dunes, river terraces and lake shorelines, as well as the sediments derived from weathering and erosion. In practice, archaeologists will seldom rely upon purely geomorphological evidence in reconstructing past environments, but such evidence is a useful start and has the great advantage that it is found in all continental environments. In addition to seeking information on the nature of the environment associated with a particular set of prehistoric or historic artefacts, the archaeologist will also need to know whether the artefacts have been disturbed after deposition or whether they remain in primary context, that is, relatively undisturbed.

Prehistoric archaeology extends back over 2.5 million years to when the first stone tools appear in the archaeological record. On the present evidence, the oldest stone tools come from the Gona Valley (Figure 1) in the now very arid western Afar Rift of Ethiopia [14,15]. (Figure 1 shows the location of places cited in the text). The tools consist of stone flakes struck from larger cobbles or pebbles. These pebble tools and flakes make up what is called the Oldowan stone tool tradition—named after Olduvai Gorge (Figure 1) in Tanzania—and are the oldest stone tool assemblage of the Lower Palaeolithic or Early Stone Age. These Oldowan tools appear at a time when the earth's climate was becoming increasingly cold and more variable, a time known to geologists as the Quaternary Period. The Quaternary spans the past 2.6 million years and was a time when great ice sheets waxed and waned in high and middle latitudes, sea levels rose and fell, rain forests expanded and contracted, tropical lakes appeared and disappeared, and deserts advanced and retreated [8,9]. These fluctuations had a profound effect upon plants and animals. Much of the evidence for these fluctuations comes from geomorphology. Prehistoric humans responded to these environmental changes in a variety of ways, including developing a wider range of tools as well as migrating to more favourable regions [16,17]. Plants, animals, and human groups unable to adapt to these changes simply vanished. Prehistoric archaeology is thus a tale of survival or extinction. This review illustrates the unique contribution geomorphology can provide to our understanding of the archaeological record.

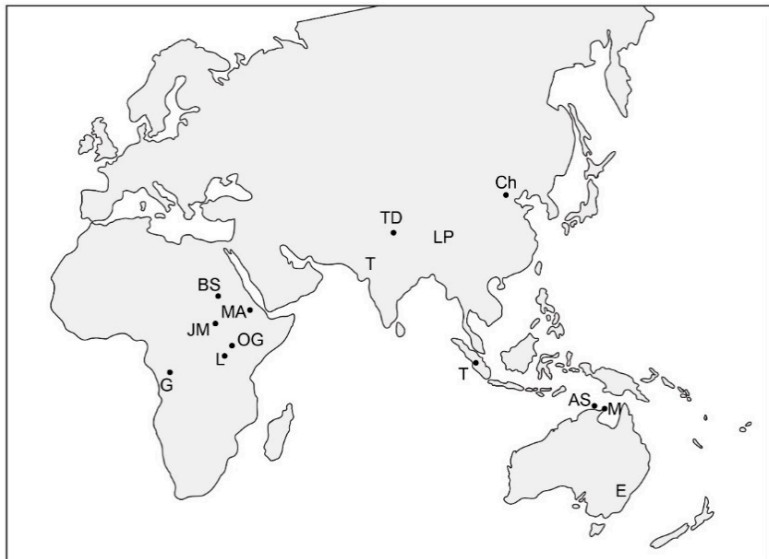

**Figure 1.** Location maps showing places cited in the text. From west to east in Africa, places are as follows: G is Gombe prehistoric site; JM is Jebel Marra volcano; BS is Bir Sahara and Bir Tarfawi; L is Laetoli; OG is Olduvai Gorge; MA is Middle Awash Valley, Afar Rift and Gona prehistoric site. From west to east in Asia, T is Thar Desert, India; TD is Taklimakan Desert, China; LP is Loess Plateau, China; T is Toba volcano, Sumatra; and Ch is Choukoutien prehistoric site, near Beijing, China. In Australia, AS is Arafura Sea, M is Millingimbi, and E is Eastern Highlands including the headwaters of the Lachlan River.

## 2. Overview

### 2.1. Geomorphological Evidence

Table 1 is adapted from [9] Table 1.1 and [11] Table 1.2. It shows the different types of geomorphological evidence commonly used in reconstructing prehistoric environments. Some of the best-preserved evidence comes from presently dry regions like the Sahara and the deserts of Arabia, Asia and the Americas, all of which have been wetter on many occasions in the past [11,17]. Deserts act as vast natural museums. Thanks to their present aridity, the evidence left from wetter times has been preserved in many favourable localities but not in areas where sand movement is active [11,17–19]. In the case of the Sahara, the

evidence for a previously wetter climate includes the remains of former lakes and rivers, with their associated fossil fauna and flora [20–30]. The lake and river sediments are often partly buried beneath wind-blown sand and are sometimes found sandwiched between layers of wind-blown sand, indicating an alternation of wetter and drier climatic phases during prehistoric times (Figure 2). Because water is essential for life, most archaeological occupation sites will occur close to the rivers, lakes, springs and wetlands that provided them with water when they were occupied. We will therefore begin by looking at the evidence provided by rivers, lakes, springs and wetlands before discussing other types of geomorphic environment. Before doing this, we need to explain how we might describe such evidence.

**Table 1.** Geomorphological evidence used to reconstruct prehistoric environments. (Adapted from [9] Table 1.1 and [11] Table 1.2).

| Type of Evidence |
| --- |
| **Rivers** |
| Channel pattern: braided, meandering, straight. |
| Channel features: banks, levees, mid-channel bars, point bars |
| Channel sediments: gravel, sand, silt, clay |
| Abandoned channels |
| Flood plains and flood plain sediments |
| Abandoned flood plains: river terraces |
| Alluvial fans |
| Pediments |
| Wetlands |
| **Lakes** |
| Reservoir and amplifier lakes |
| Lake shorelines and associated beach ridges |
| Lake sediments |
| Deltas |
| **Ice** |
| Glacial deposits: glacial moraines |
| Glacial erosion: glacial valleys, glacial cirques. |
| Periglacial deposits: angular rubble, block streams, solifluction mantles |
| Glacial loess |
| **Wind** |
| Desert dust |
| Dunes and sand plains |
| Clay dunes |
| Source-bordering dunes |
| **Coasts** |
| Coastal beach barriers and lagoons |
| Coastal plains |
| Chenier plains |
| Deltas |
| **Volcanoes** |
| Volcanic ash |
| Lava flows |

*2.2. Describing the Evidence*

Although in some cases archaeological remains may simply lie on top of a bare gravel or rock surface (Figure A1), in most cases the remains are found embedded within sediments. Some of these sediments may have developed subsequently into soils. Soil is weathered rock and sediment in which plants grow. Together with water and air, soil is essential to life on this planet. The presence of buried soil within alluvial sediments (Figure 3) indicates that there was a break in sediment deposition and that the sediment was exposed to plant growth and associated biological activity and other soil-forming processes.

Both the soils and the sediments can tell us a great deal about the type of environment in which the prehistoric artefacts were first deposited.

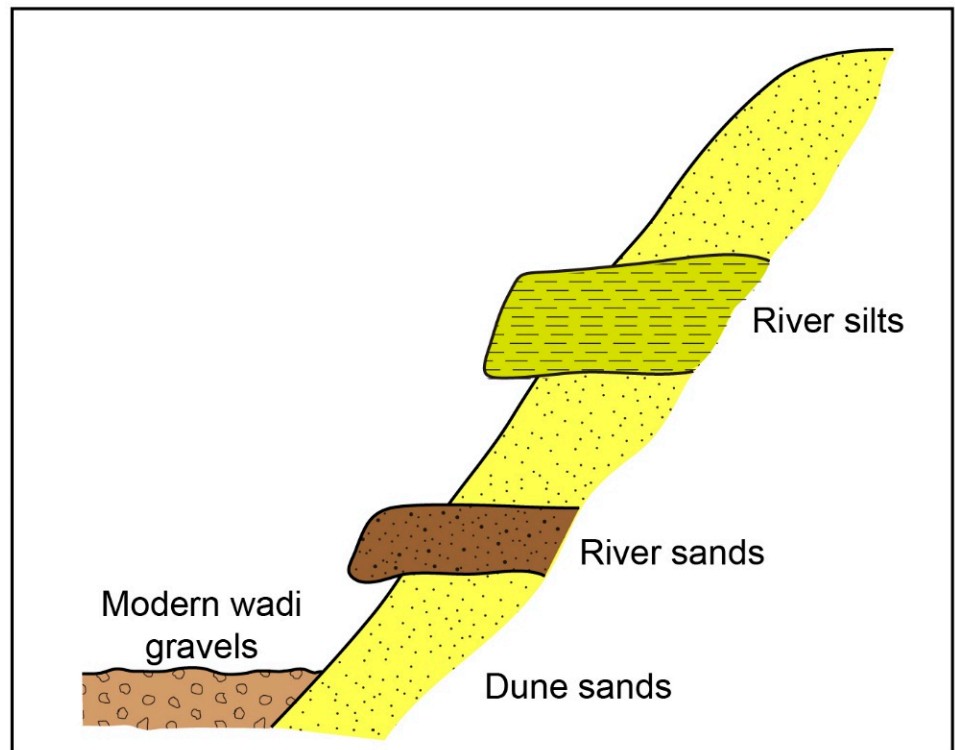

**Figure 2.** Alternating wind-blown sands and river and lake sediments, indicating alternating wetter and drier climatic conditions. Based on the author's observations in the south-central Sahara.

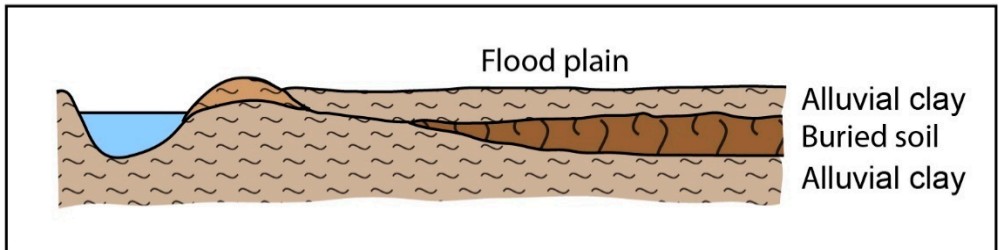

**Figure 3.** Buried soil within stratified alluvial flood plain sediments, indicating a break in the deposition of flood plain sediments long enough for a soil to develop. Based on the author's observations immediately east of the Nile in northern Sudan.

When describing the soils and sediments associated with archaeological remains, earth scientists generally use a standard method of description. Soil scientists tend to describe soils in very much the same way as the detailed passages in various editions of the Soil Survey Manual of the U.S. Department of Agriculture (e.g., [31,32]) with minor national differences to account for regional variations in soils. Geologists around the world have also adopted a relatively uniform set of terms and descriptions for sediments and sedimentary structures (see, for example, Chapter One in [33]), but these terms will always be in the process of revision as new information comes to light. Table 2 shows some of the more important attributes of soils and sediments that may be recorded in the field and followed up on with certain laboratory analyses. Such analyses are generally determined by the type of questions being asked, such as how old a piece of charcoal is or how well-sorted a particular sediment sample is.

**Table 2.** Describing soils and sediments. (Based on [31,32], supplemented by the author's field observations since 1962 in Africa, Asia and Australia).

| Field Observations |
| --- |
| Thickness of each stratigraphic unit; depth below surface |
| Nature of its contact with adjacent units |
| Munsell Chart colour (both wet and dry) |
| Soil structure, soil consistence and hardness, |
| Field texture (to within about 5% clay content, for sixteen soil textural classes) |
| Sand particle shape and degree of roundness (under $\times 10$ and $\times 20$ magnification) |
| Sedimentary structures |
| Clasts (size, shape, composition, %) |
| Carbonate concretions (shape, size and %) |
| Hardpan layers |
| Calcrete horizons |
| Silcrete horizons |
| Ferricrete (ironstone) horizons |
| Gypsum (crystals, powdery) |
| Micro-crystalline calcite and/or dolomite |
| Presence of snail shells (aquatic, semi-aquatic, terrestrial), |
| Charcoal fragments or other organic material. |
| Bones |
| Archaeological material (e.g., prehistoric stone artefacts, hearths, pottery, middens, burials) |
| Where appropriate, details of samples collected for OSL and/or $^{14}$C analysis. |
| **Laboratory analyses** |
| Particle size |
| Micro-morphology |
| Light and heavy minerals |
| Isotopic analyses (e.g., carbon, oxygen, deuterium, strontium, neodymium) |
| Chemical properties of the soils, including salinity (electrical conductivity) and alkalinity (pH; exchangeable sodium percentage) |
| Pollen analysis |
| Shell identification |
| Fossil vertebrate and invertebrate analysis (macro-fossils and micro-fossils) |

*2.3. Evidence from Rivers*

From the time that the earliest hominins first appeared in the Middle Awash Valley (Figure 1) of the Ethiopian Afar Rift (Figure 1) over five million years ago [34,35], humans and their remote ancestors have always sought the security of being close to water, as discussed earlier. Big rivers like the Nile that flow through deserts have attracted humans since early prehistoric times [16]. Changes in the location of river channels through time will be reflected in corresponding changes in the location of prehistoric occupation sites, as in northern Sudan, where former channels of the Nile are very well preserved [36].

River channels are very dynamic features. They can alter their shape in response to changes in sea level, earth movements and the supply of water and sediment from their headwaters [37–40]. Rivers that carry mainly sand and gravel tend to be wide and shallow, with numerous mid-channel bars of sand and gravel (Figure 4a). Such river channels are called braided channels (Figure 4a) and are common in deserts and in glaciated regions on moderately steep slopes. They tend to follow a relatively straight course and are also known as bedload rivers. In contrast, a river channel which is transporting mostly very fine silt- and clay-sized particles in suspension is relatively deep and narrow and often follows a very sinuous or meandering course (Figure 4b). Such rivers are called suspension load rivers and usually flow from well-vegetated upper catchments with deep soils across gentle slopes. Changes in channel patterns over time can tell us about changes in the headwaters. For example, if a river that was previously wide, shallow and braided becomes deep, narrow and meandering, this may indicate that the previously dry climate in the headwaters has altered to a wet climate, leading to more luxuriant vegetation and to enhanced weathering

and soil formation. Two case studies, one from Africa and one from India, will serve to illustrate the complex response of big rivers to Quaternary climatic fluctuations.

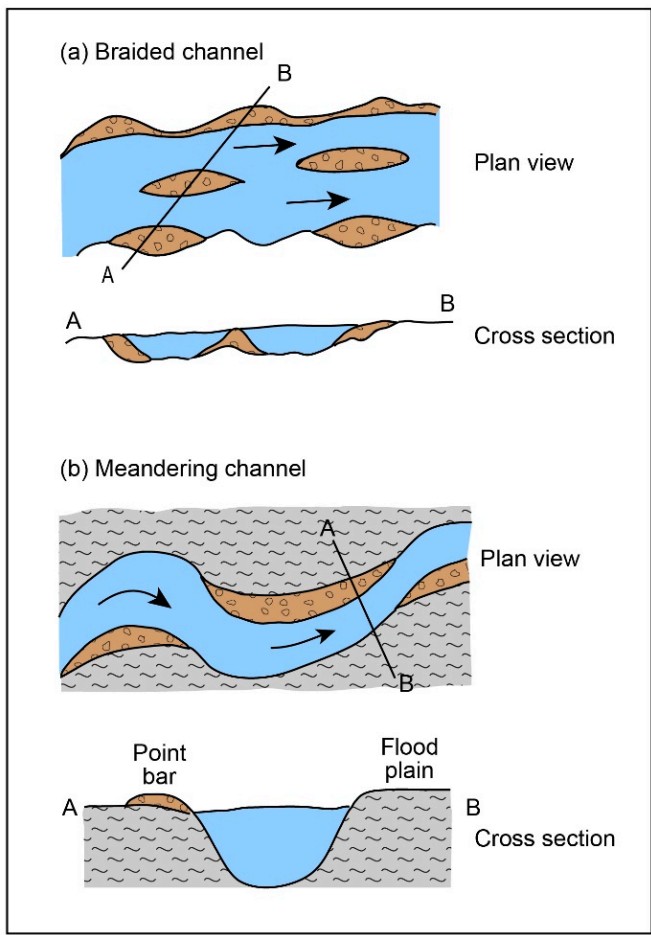

**Figure 4.** Channel plan form and cross section. (**a**) Braided channel. (**b**) Meandering channel. (**a**) is based on the author's observations in the South Island of New Zealand; (**b**) is based on the author's observation east of the lower White Nile, central Sudan.

2.3.1. Late Quaternary Alluvial History and Prehistoric Environments in the Nile Basin

In the case of the Ethiopian headwaters of the Nile, notably the Blue Nile and Atbara rivers (Figure 5), the regional climate has alternated between cold and dry and warm and wet during the last few million years. For example, during the Last Glacial Maximum, some 20,000 years ago, the climate in the Nile headwaters was colder and drier than it is today [16,41,42]. Plant cover was sparse, and frost action was widespread above an elevation of about 3000 m [43,44]. The hill slopes were unstable and released huge volumes of coarse sediment into the Blue Nile and Atbara, which carried a bed load of coarse sand and gravel (Figure A2) which they deposited in northern Sudan and southern Egypt during the dry winter months at a time when flow in the main Nile had dwindled to a trickle or ceased altogether [45]. The reason for this much-reduced winter flow is found in the Ugandan headwaters of the White Nile (Figure 5), which dried out at that time [46–49]. Today, the White Nile provides 85% of the low-season flow, and before dams were built, this river ensured perennial flow in the main Nile during dry years. During the Last Glacial Maximum, the large lakes in Uganda that normally fed the White Nile dried up so that flow in that river effectively ceased [16]. As a result, the main Nile in Egypt and northern Sudan dried up in winter, and the Upper Palaeolithic people in those areas were forced to live around isolated ponds impounded between desert dunes [45].

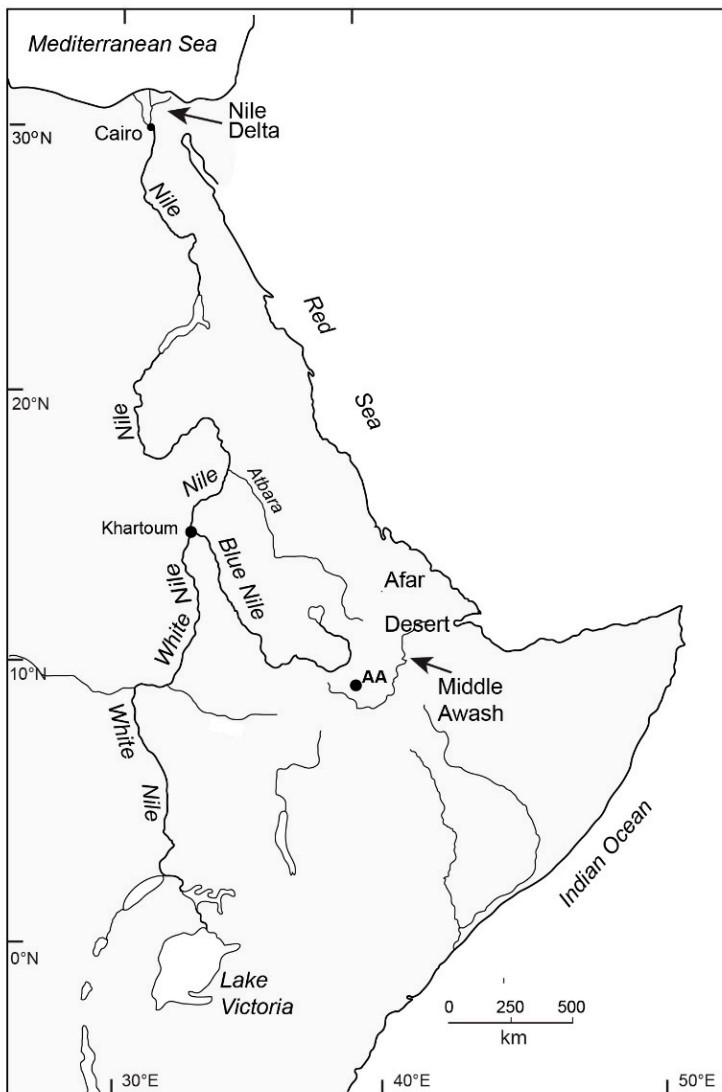

**Figure 5.** Nile basin location map. AA is Addis Ababa, capital of Ethiopia.

Once the African summer monsoon became re-established soon after 15,000 years ago [50,51], the flow regime of the Nile changed dramatically. The White Nile resumed flowing, ensuring that the main Nile returned once more to a perennial flow regime. The flood season in the Blue Nile and Atbara rivers now lasted longer, and the volume of water flowing to the main Nile and on to the Nile Delta (Figure 5) was greater than before. However, the most significant change was the type of sediment transported by the Nile, which now ferried a large volume of silt and clay [16,41]. During the summer months, these fine alluvial sediments helped to build up the flood plains along the Nile, so that each year, soil fertility was replenished.

Understanding regional changes in prehistoric environments provides the archaeologist with a useful general background but one needing to be supplemented by more detailed information at the level of the individual site. Table 3 gives a summary of the type of local geomorphic evidence that has been used to reconstruct the environments associated with Mesolithic and Neolithic sites in northern Sudan and the lower White Nile valley [52,53]. The geological term facies refers to the characteristics of a sedimentary deposit. It includes the grain size, bedding, mineral composition and fossil content, all of which reflect the depositional environment (see [53] for discussion). Figure 6 shows a hypothetical cross section from the main Nile channel across its flood plain to where it merges with local wadi sediments.

**Table 3.** Depositional environments and associated sedimentary facies in the Nile valley. For further details, see [52–54].

| Description | Interpretation |
|---|---|
| Horizontal beds of clay or very fine sandy clay, sometimes separated by beds of fluvial sand or silt but sometimes stacked one above the other in beds up to a metre thick, with occasional fine laminations visible within the beds. | Flood plain deposits laid down by the river during the summer flood season. |
| Horizontal beds of fine fluvial sand, showing planar and cross-bedding sedimentary structures. Bed thickness varies from a few centimetres to a metre. In a few cases bands of sand and silt a few centimetres thick alternate with more clay-rich bands of similar thickness. | Bed load channel sediments or overbank deposits laid down during exceptional floods. The alternating fine sand and clayey units are probably levee deposits that were laid down adjacent to former river channels |
| Planar- and cross-bedded gravels and coarse sandy gravels. The gravels consist of sub-rounded quartz and carbonate pebbles 1–5 cm in diameter set within a matrix of coarse sand and fine granules. | The quartz and carbonate gravels represent bed load gravels transported during phases of very high-energy flow and highly seasonal discharge. More local gravel bands reflect sporadic local flood events |
| Calcareous silts, silt loams, clay loams and marls occupying shallow pans, with abundant shells of aquatic, semi-aquatic and terrestrial snails. Colours vary from white to grey to olive brown. | Shallow ephemeral or seasonal pans filled during times of wetter seasonal climate. |
| Massive fine sandy clays with sporadic aquatic and semi-aquatic shells | Wetland deposits laid down in relatively large seasonally flooded depressions. |
| Wind-blown medium to fine well-sorted sands with planar bedding and cross-bedding sedimentary structures, and a heavy mineral suite characteristic of the alluvial sands from which they were derived. | Source-bordering dunes derived from sandy alluvium transported by highly seasonal streams. |

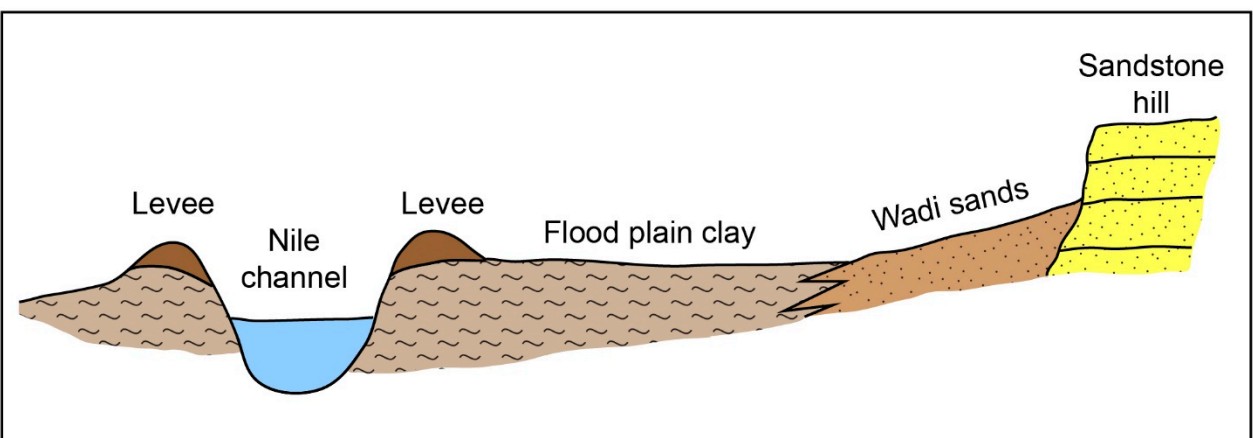

**Figure 6.** Hypothetical section from the main Nile channel across its flood plain to where it merges with local wadi sediments. Based on the author's observations immediately east of the Nile in northern Sudan.

2.3.2. Alluvial History and Prehistoric Environments in the Middle Son Valley, North-Central India

Early archaeologists working in Europe and Asia were often unduly focussed on searching for prehistoric fossils and stone artefacts within river gravels exposed in the banks and alluvial terraces of modern rivers. In the Belan valley of north-central India (Figure 7), the apparent association between Lower, Middle and Upper Palaeolithic stone artefacts and three alluvial gravel beds had misled archaeologists into believing that the artefacts were in situ until a new approach based on detailed stratigraphic mapping was adopted from 1980 onwards [55–60]. Any artefacts found within river gravels must have been transported and reworked and are either about the same age as the gravel or older. River gravels can therefore carry and deposit artefacts and fossils of widely different ages.

Knowing when a gravel was deposited does not provide an age for the artefacts and fossils laid down with the gravel. Alluvial gravel beds also tend to be discontinuous. What is needed to resolve this impasse is to identify individual alluvial formations by careful mapping on the ground and to recognise that alluvial formations are three-dimensional bodies of sediment which can vary vertically and laterally.

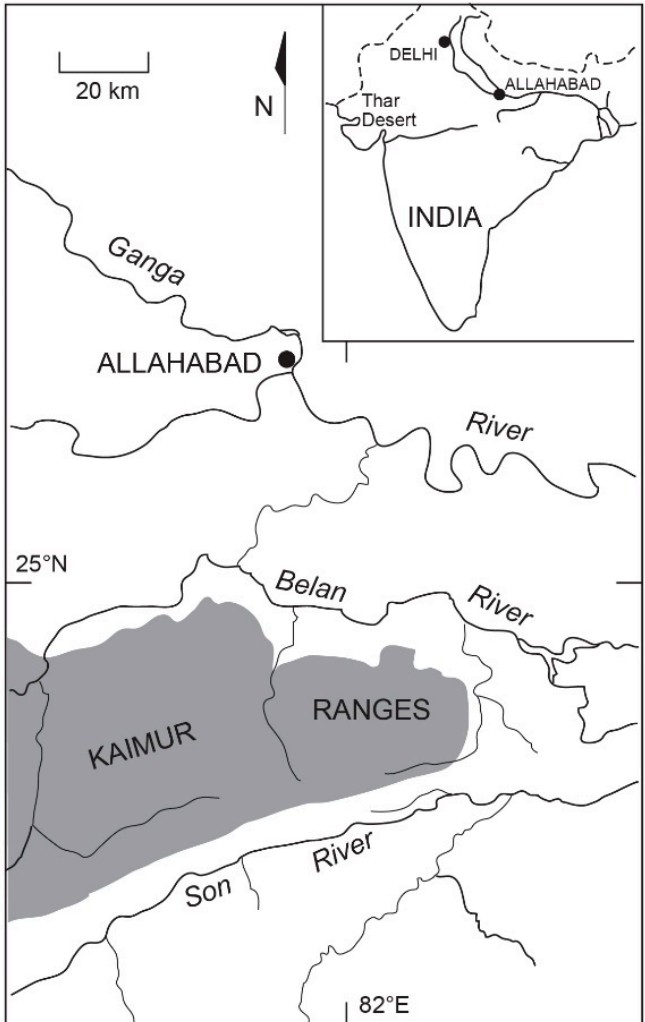

**Figure 7.** Location of Son and Belan Rivers in north-central India.

Table 4 lists the main characteristics of the five main alluvial formations that have been identified so far [61] in the Middle Son valley (Figure 7) close to the headwaters of the Belan River mentioned earlier. The formation names remain informal. Careful mapping in the field by earth scientists has allowed archaeologists to identify those environments in which the artefacts were most likely to be in primary context and those where they had been transported [57,61]. For instance, the Baghor Formation consists of two alluvial units or members: a coarse-grained Lower Member and a fine-grained Upper Member (Table 4 and Figure 8a,b). The Lower Member consists of cross-bedded coarse sands and gravels and was laid down under conditions of high-energy flow (Figure A3). The abundant vertebrate fossils within this unit have been transported and are not in primary context. In contrast, the fine-grained sediments that make up the Upper Member of the Baghor Formation were laid down as flood plain clays under conditions of very low-energy flow and contain a wealth of undisturbed archaeological evidence. There are even well-preserved prehistoric tracks of sambur deer on the surface of a clay layer which was laid down on the former flood plain some 8000 years ago (Figure A4).

**Table 4.** Interpreting depositional environments in the Middle Son valley, north-central India. (Modified and updated from [61]).

| Formation Name (Provisional) | Interpretation |
| --- | --- |
| **Sihawal Formation**<br>**Lower Member**: poorly sorted angular quartzite gravel in silty clay matrix with sporadic fresh Lower Palaeolithic bifaces.<br>**Upper Member**: massive, grey silty clay entirely devoid of artefacts and sedimentary structures. | Local alluvial fan deposits contemporary with primary context Lower Palaeolithic artefacts.<br>Wind-blown dust or loess, possibly locally reworked by runoff and soil creep. Dry climate. |
| **Khunteli Formation**<br>Upper Pleistocene fluvial sands and gravels with Middle Palaeolithic artefacts overlying a bed of relatively pure volcanic ash. Abundant calcium carbonate concretions throughout the deposit. | Alluvial sediments laid down by a seasonal bed-load river. Precipitation of calcium carbonate consistent with a dry semi-arid climate during or soon after deposition. The volcanic ash came from Toba volcano in Sumatra 74,000 years ago. |
| **Patpara Formation**<br>Fining-upwards red and reddish-brown alluvial gravels, sands and clays with abundant agate, chalcedony and jasper pebbles in the lower gravel members. The Middle Palaeolithic artefacts range from highly abraded and transported to retaining sharp edges indicative of a nearby source. | Alluvium derived from source in Deccan Traps near headwaters of Son River. The sediments have been extensively weathered, with precipitation of iron under oxidising conditions during a time of wetter-than-present regional climate. |
| **Baghor Formation**<br>**Lower Member**: cross-bedded and planar-bedded coarse sands and fine gravels, with abundant transported and carbonate-cemented vertebrate fossils. Abraded and transported Upper Palaeolithic artefacts.<br>**Upper Member**: horizontally bedded alluvial clays, with primary context Mesolithic and early Neolithic artefacts. | Sparse plant cover, rapid hillslope erosion and active sedimentation in the valley. Weaker summer monsoons; colder, drier climate.<br>Terminal Pleistocene/early Holocene flood plain deposition. Wetter regional climate. |
| **Khetaunhi Formation**<br>Horizontally bedded late Holocene clays, silts and sands, now forming an alluvial terrace close to present-day river level. | A brief interval of fine-grained floodplain sedimentation during a short wetter climatic interval in the late Holocene followed by a return to a more seasonal climate and limited channel incision. |

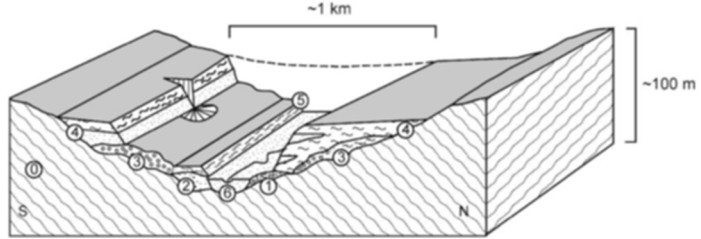

⓪ Lower Proterozoic metasediments
① Middle Pleistocene gravels and clays of the Sihawal Formation
② Upper Pleistocene sands and gravels of the Khunteli Foramtion
③ Upper Pleistocene gravels, sands and clays of the Patpara Formation
④ Terminal Pleistocene sands and clays of the Baghor Formation
⑤ Late Holocene clays, silts and sands of the Khetaunhi Formation
⑥ Present-day channel sands and point-bars of the River Son

(**a**)　　　　　　　　　　　　　　　　　　　　(**b**)

**Figure 8.** (**a**) Block diagram showing the five major alluvial formations investigated by the author in the Middle Son valley. (After [61] Figure 3 and [62] Figure 2). (**b**) Schematic section of the Baghor Formation in the Middle Son valley of north-central India showing the coarse-grained Lower Member and the fine-grained Upper Member. (For details, see [55,61]).

*2.4. Evidence from Lakes, Springs, and Wetlands*

Lakes, like rivers, have always attracted prehistoric humans, particularly in places where the climate was seasonally dry, as in many parts the tropics. At two localities called Bir Sahara and Bir Tarfawi (Figure 1) in the now hyper-arid eastern Sahara, there was a series of lakes associated with Middle Palaeolithic sites which date back to warm, wet

Last Interglacial times 125,000 years ago [63]. It is possible that there were links between these lakes and the Nile at that time because the fossil fish fauna is like that found in the Nile today [64]. The local groundwater table would also have been higher then [63]. The clays and silts that accumulated in the centres of these lakes survived the ravages of sandstorms after the lakes had dried up, but the sandier lake shorelines have long since been blown away by strong winds. Where prehistoric lake shorelines are reasonably well preserved, and there is some independent control over temperature and evaporation, it may be possible to model former precipitation onto the lake and to construct a hydrologic budget [65,66]. However, the results of such an exercise will always need to be tested against independent evidence from fossil pollen spectra and/or from the stable isotopic composition of aquatic snail shells.

Certain lakes are highly sensitive to local changes in rainfall and runoff, particularly if the catchment area is large relative to the size of lake. Such lakes are called amplifier lakes [67]. Not all lakes are sensitive to climatic fluctuations, notably lakes that are fed primarily from groundwater or lakes that are simply enlarged portions of a river flowing into and out of these lakes. These latter lakes can be considered reservoir lakes [67]. The Willandra lakes in arid western New South Wales, Australia, including Pleistocene Lake Mungo (Figure 9) [68], are reservoir lakes that once received water from a former distributary channel of the Lachlan River which rises in the Eastern Highlands of Australia. The fluctuations in the level of Lake Mungo reflect changes in runoff from the upland headwaters of the Lachlan and have little to do with local climatic fluctuations.

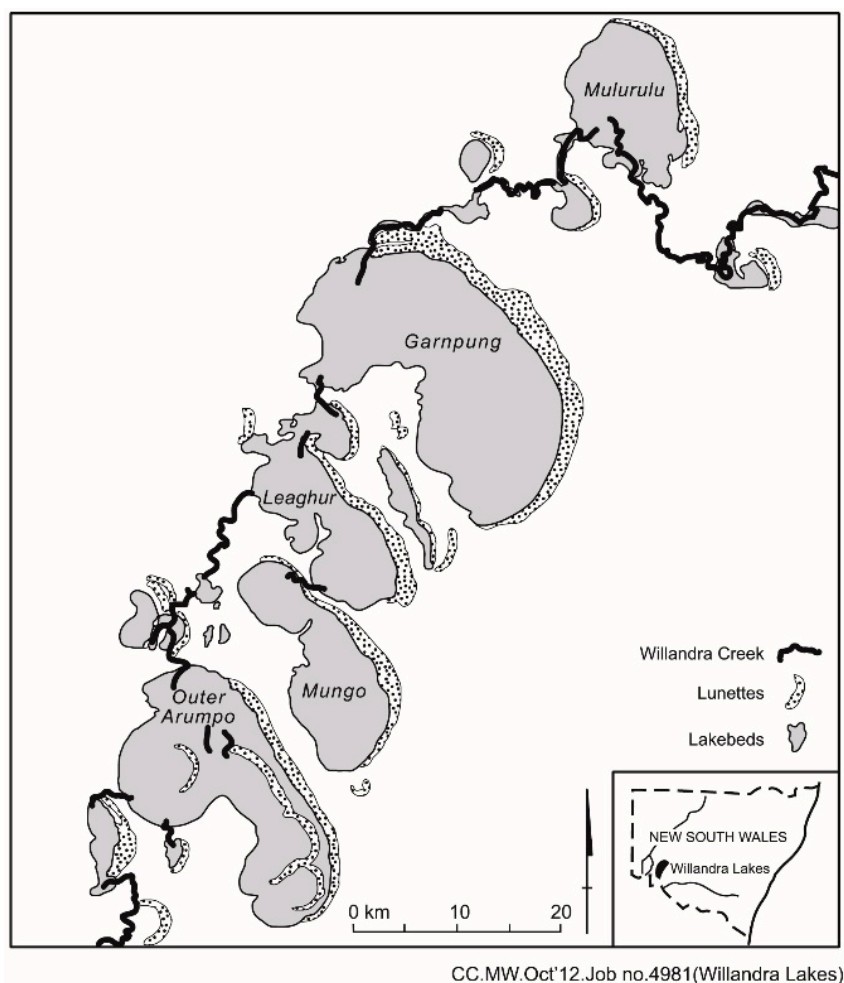

CC.MW.Oct'12.Job no.4981(Willandra Lakes)

**Figure 9.** The late Pleistocene Willandra lakes in semi-arid New South Wales, Australia showing Lake Mungo. (After [11] Figure 11.3).

During the wet climatic phase between about 14,500 and 5000–4500 years ago, much of the Sahara was studded with lakes large and small [18,69–74]. These lakes were attractive to small groups of Upper Palaeolithic hunter–fisher–gatherers. The middens of these people also contain the bones of hippos, crocodiles, turtles and wild fowl [75,76]. Some of the lakes dried out about 8500 years ago and refilled to a lower level shortly before 6300 years ago, when Neolithic cattle herders moved into the Sahara [77–80]. They remained there until aridity set in about 5000 years ago and forced them to move to areas where water was available all year, including the Nile valley [17].

In the Afar Desert of Ethiopia (Figure 1), the margins of former lakes contain fossil remains and stone tools dating back at least 2.5 million years [15], while hominin fossils dating back to over 5 million years ago [34,35] also occur in localities where rivers once flowed into lakes as well as on former flood plains and along former lake shorelines. In the Ethiopian and Kenyan Rift valleys, where large, deep lakes are very common, there is a rich record of prehistoric human occupation extending back to Lower Palaeolithic times. The same is true of Olduvai Gorge (Figure 1) in Tanzania, with its complex record of river and lake sediments, volcanic ash beds, and fossil soils [81,82].

Springs were another source of permanent water for prehistoric humans. Springs were often the only source of fresh water in desert regions such as central Australia, the eastern Sahara, and the Taklimakan Desert (Figure 1) in western China. The springs ceased flowing once prolonged aridity caused the regional water table to fall. Deposits of calcium carbonate called *tufa* mark the position of the vents of these prehistoric springs. Embedded within the tufa there may be stone artefacts of varying ages and showing different degrees of disturbance, and so undue reliance on tufa deposits to reconstruct former patterns of prehistoric occupation is seldom wise. However, the types of artefacts preserved within the tufa (for example, Mousterian or Aterian) can provide a very rough indication of when the area was occupied by prehistoric people.

Wetlands were also a source of water and aquatic food resources for prehistoric people. Even today, during times of drought the women in the Afar Desert of Ethiopia collect water lily bulbs from the surviving wetlands and grind the small black seeds inside the bulbs to make a type of porridge that is a valued form of famine food.

## 2.5. Evidence from Dunes and Desert Dust

During times of prolonged aridity, desert dunes are at their most mobile, such that even if they were occupied, evidence of occupation is unlikely to survive. Bare mobile sand dunes far from water and devoid of vegetation would in any case be unattractive sites for anything other than transitory use. Once wetter conditions resumed, plants would colonise the dunes, soils would form and stabilise the previously highly mobile sand, and the dunes would become more attractive sites for seasonal or permanent occupation.

In certain situations, sand dunes have been attractive occupation sites for prehistoric people. In the lower White Nile valley (Figure 5), the summits of sand dunes were occupied at least seasonally by Mesolithic hunter–gatherers during a time of wetter climate [83]. Shell middens are common on the dunes in this region (Figure A5). The Mesolithic people collected the large semi-aquatic *Pila* snails from the seasonally flooded wetlands surrounding the dunes and either used them for bait when fishing for Nile perch or used them for food, most likely after boiling the snails in clay pots [83].

Dunes can sometimes contain a very long record of prehistoric occupation. In the Thar Desert (Figures 1 and 7) of northwest India, the dunes consist of alternating layers of wind-blown sand and buried soils. These soils are sometimes cemented with calcium carbonate to such an extent that they have successfully resisted erosion. One such site extends back over 200,000 years [84] and consists of twelve fossil soils alternating with wind-blown sands (Figures 10 and A6). The soils represent wetter climatic intervals and contain prehistoric stone artefacts extending back in time from the Mesolithic through the Middle Palaeolithic to the final stages of the Lower Palaeolithic.

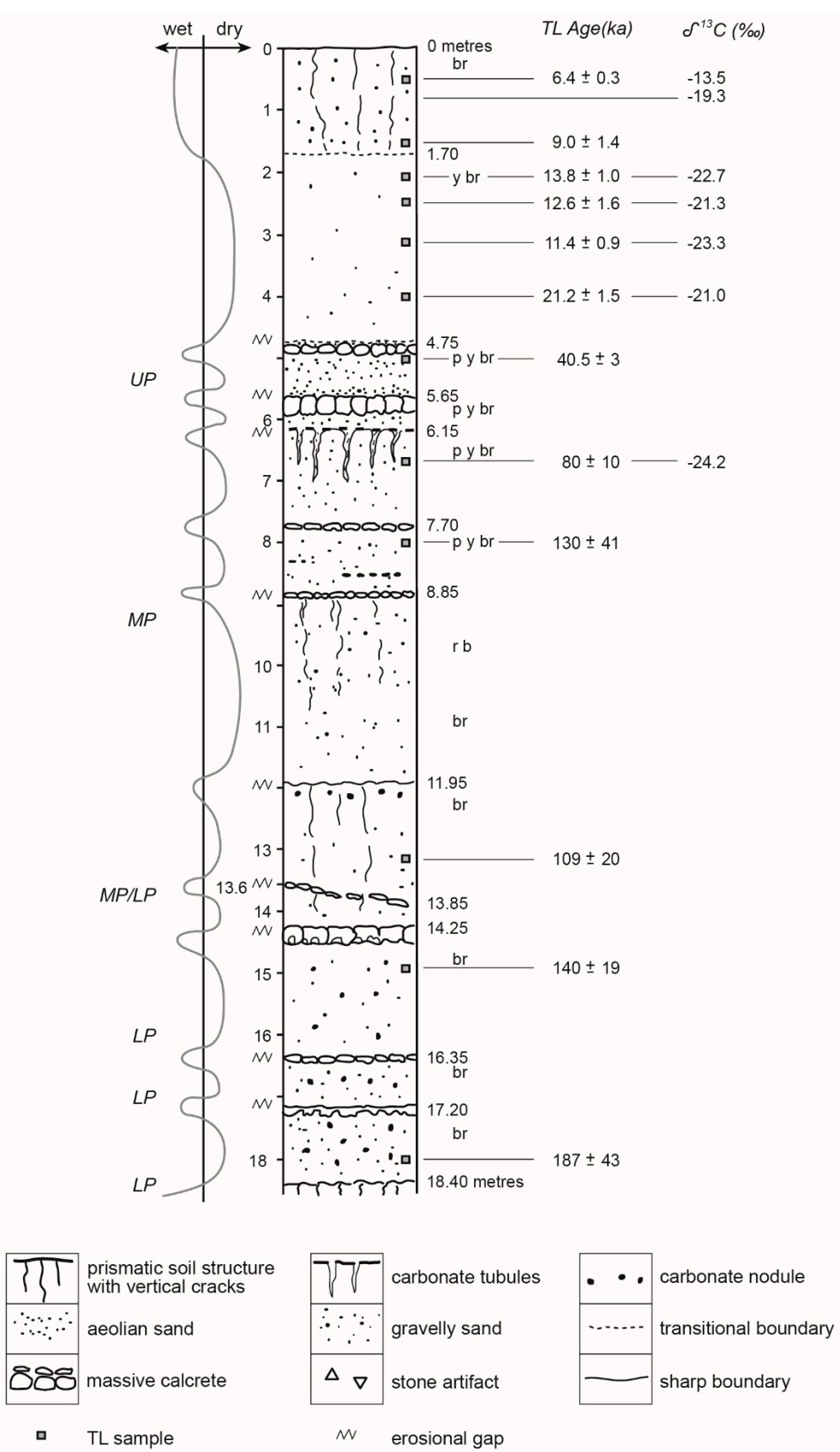

**Figure 10.** Stratigraphic section through a Quaternary polygenic dune in the Thar Desert, India, showing eleven alternating phases of soil/calcrete formation and sand accretion during the last 200, 000 years. (After [84] Figure 3 and [11] Figure 8.11).

On a far grander spatial and temporal scale, the Loess Plateau of China (Figure 1) contains an unrivalled record of alternating fossil soils and wind-blown desert dust or *loess* [85–88]. The Chinese loess record spans several million years. The soils represent times of stronger summer monsoons and generally wetter climate [89], with the most recent wet phase coinciding with the spread of Neolithic rice cultivation across southern China [90]. The loess layers represent times of rapid accumulation of desert dust under a climate that was cold, dry and windy. Such conditions would have made life difficult out on the open plains and would have encouraged people to seek refuge in limestone caverns during the long winter months. The cave site at Choukoutien near Beijing (Figure 1) shows evidence of prolonged occupation [9] and sustained use of fire in the two main occupation levels, with the fossil plant and vertebrate remains found in the thick ash layers consistent with very cold outside temperatures in the vicinity of the caves.

Clay dunes or *lunettes* are a very particular type of dune that is found on the downwind margins of seasonally fluctuating lakes [91,92] like Pleistocene Lake Mungo in Australia (Figures 9 and A7). The clay components of these dunes were blown from mud curls exposed near the centre of the former lake during times of minimum lake level and are present in the form of clay pellets that look like sand and behave like sand but are in fact aggregates of clay. The sandier beds within lunettes were blown from sandy beaches and are generally indicative of higher lake levels. The lunette at Pleistocene Lake Mungo has revealed human burials as well as remains of aquatic and terrestrial animals from the long interval of time between about 45,000 and 20,000 years ago when humans lived around the lake [68].

### 2.6. Evidence from Volcanoes

The initial impact of a volcanic eruption is destruction, with the blast and the red-hot lavas destroying the plant cover [93] and even altering the shape of the landscape. Depending on the magnitude of the eruption, great clouds of volcanic ash can be ejected high into the atmosphere and can even circle the Earth for several years. The ash particles eventually settle upon the surface of the Earth, where they can form layers of volcanic ash, or *tephra*, many centimetres thick. Certain types of ash become very hard soon after they are deposited on the surface of the ground, and some have preserved prehistoric tracks such as the hominin footprints at Laetoli (Figure 1) in Tanzania, which are about 3.7 million years old [94].

Ash that erupted from the Toba volcano (Figure 1) in Sumatra 74,000 years ago reached as far as India, which was covered in a layer of ash 10–15 cm thick known as the Youngest Toba Tephra, or YTT [95,96]. The impact of this eruption upon the regional and global climate is still hotly debated, with some arguing for little or no impact and others for a significant impact [97–101]. If an ash layer remains undisturbed over time, it can provide a very useful marker bed against which to assess whether there were any changes in the archaeological and environmental records following deposition of the ash.

In the case of India, the YTT has the potential to provide a time marker enabling prehistoric archaeologists and earth scientists to compare the nature of the environmental and Palaeolithic record before and after the eruption. To do this, it is essential to show that the ash bed is undisturbed. In most cases, the ash will be subject to erosion and will be deposited lower in the landscape. Even ash that was transported downslope and buried beneath younger sediment soon afterwards can be a useful marker bed.

Volcanic ash can also help preserve plant fossils and artefacts. Ash from the eruption of the Jebel Marra volcano (Figure 1) in arid western Sudan contains leaf fossils of the oil palm *Elaeis guineensis* (Figure A8) as well as stone tools dating back about a million years. The oil palm fossils show that the climate was much wetter at that time [16].

### 2.7. Evidence from Coasts

As the great northern ice caps built up progressively during each Quaternary glacial cycle, so much water was locked up in the form of ice that global sea levels fell, and the previously submerged continental shelves became dry land. Plants and animals gradually colonised these newly exposed lands, as did groups of prehistoric humans. One preferred

habitat for prehistoric humans was the coast, because this allowed access to both terrestrial and marine resources. The most recent time of very low sea level was during the Last Glacial Maximum about 20,000 years ago, when sea levels fell by about 125 m. Where the continental shelves were gently sloping, vast tracts of land were exposed, as in southeast Asia and around Australia. Once the ice melted, and the sea level rose once more, huge areas of land became submerged [102,103], and coastal dwellers were compelled to move inland or travel by boat to other lands. In some instances, the loss of land amounted to about a metre per fortnight, which would have required a rapid and flexible response.

It was not simply a case of land lost; it was also the loss of certain coastal resources. For example, in tropical northern Australia the mangrove swamps are a rich source of marine food, including large shellfish, crabs and fish. Immediately adjoining the mangrove forest, there are narrow bands of Pandanus palms. These trees are located on curved ridges of shelly debris called cheniers (Figure 11a). Cheniers form during extreme cyclones which severely damage the mangroves [104]. These extreme cyclones are rare and only recur about every few centuries in this region. The shell grit formed by storm wave action forms a beach overlying marine clay (Figure 11b). Once the mangroves recover and trap fine sediment, the shoreline advances seawards, and the former shelly beaches are left stranded inland from the coast. During the dry season, fresh water is trapped in the shell bed above the impermeable clay. These have been dry season camping sites during at least the last 5000 years. What is especially interesting is that on one recent occasion when some of the Aboriginal clan elders of Millingimbi in northern Australia (Figure 1) were out fishing in the shallow waters of the Arafura Sea (Figure 1), they pointed casually down and commented that they had once lived there. The sea level in this region attained its present level between 7000 and 6000 years ago [105]. It is possible that the elders were referring to now submerged chenier ridges which would have been a source of fresh water during the dry season even then.

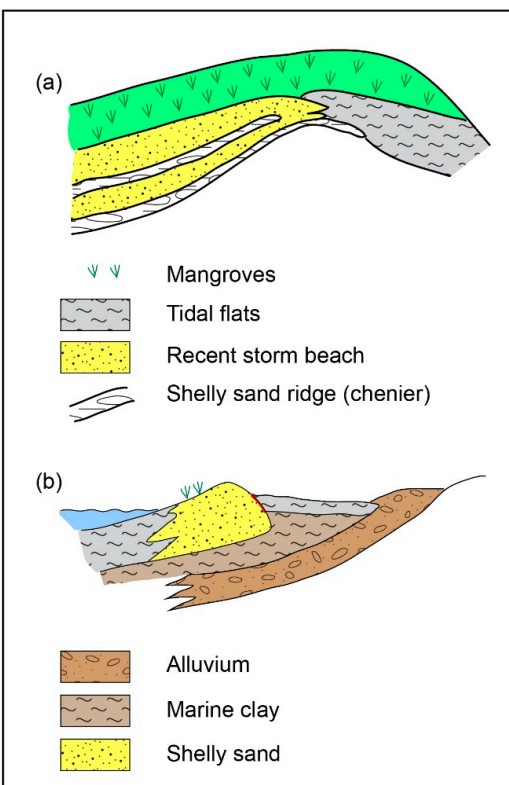

**Figure 11.** Chenier ridges in tropical northern Australia. (**a**) Transect across a chenier plain. (Adapted and greatly simplified from [104] Figure 1 and [105] Figure 6). (**b**) Generalised stratigraphy of a mangrove coast in the late Holocene, following the postglacial rise of the sea level to its present elevation at this hypothetical site some 6000 years ago. (Adapted and simplified from [105] Figure 6).

Deltas form where rivers flow into large lakes or the sea. Their fertile and well-watered soils and luxuriant aquatic and terrestrial vegetation have always made them attractive for human occupation, particularly from Neolithic times onwards [106]. Depending on when the postglacially rising sea reached its present level, much of the evidence for early human occupation is now submerged. We are therefore unlikely to find much in the way of pre-Neolithic occupation in coastal deltas. Around the coast of China, only the more recent Neolithic sites are found associated with deltas, with older Neolithic sites either further inland or beneath the sea [102].

## 3. Key Issues

*Identifying Disturbance in Archaeological Sites*

A perennial problem in archaeology is how to recognise when a site has been disturbed. In the context of prehistory, the discipline of taphonomy deals with how organisms become fossilized and preserved in archaeological sites. Initially, the focus was primarily upon fossil bones [107–110]. More recently the attention of many workers has shifted to include the role of plants [111] and insects, especially ants [112] and termites [113–117].

Related to this is the thorny question of whether ages obtained for the sediments within which artefacts occur are reliable indicators of the ages of artefacts within those sediments [5,112,118–121]. If stone artefacts are concentrated on the surface in a region prone to erosion by wind and water, it is prudent to assume that the artefacts may have been concentrated through removal of the sediments in which they once occurred. This is a common phenomenon in arid areas.

Another widespread form of disturbance is caused by the expansion and contraction of certain clay soils known as vertisols or cracking clay. Such clays are common across the semi-arid world [122–125]. In Australia the topographic features formed at the surface of such clays are known as gilgai structures. When a soil layer is overlain by a younger, denser soil, differential pressure can cause portions of the lower soil to be forced upwards through the surface soil to form hummocky features known to geologists as diapiric structures (Figure 12). Artefacts and fossil bones can be displaced upwards by such movements and will often be hard to date.

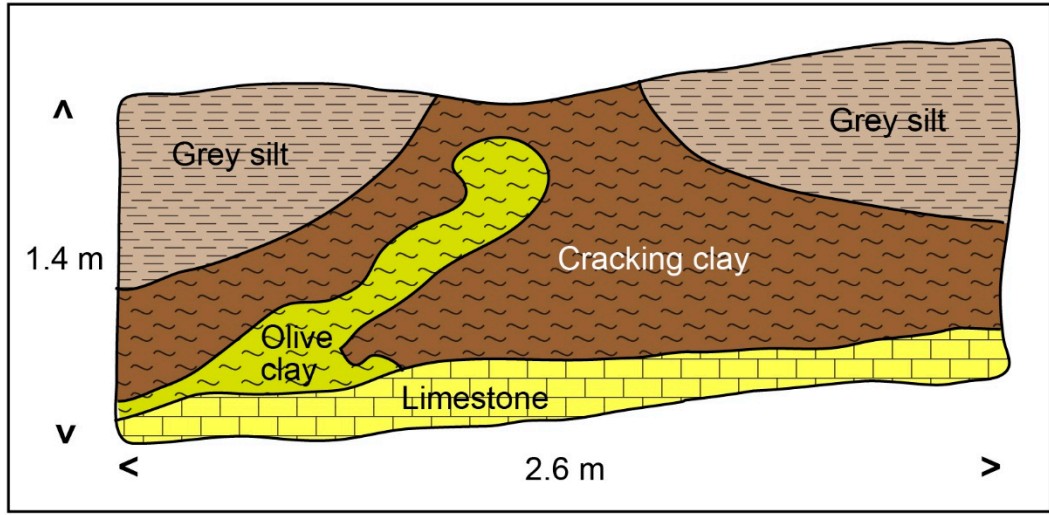

**Figure 12.** Diapiric structure in a clay soil at a prehistoric archaeological site in central New South Wales, Australia. Based on the author's observations.

It is commonly assumed that rock shelter sites are insulated from disturbance. This is not necessarily true. For example, stone layers considered to be undisturbed concentrations of stone artefacts may in fact represent the influence of termite activity [116]. At the Gombe site in Zaire (Figure 1), a combination of termite activity and soil creep has caused the downward displacement of the stone artefacts into a basal layer consisting of

artefacts derived from what were initially discrete and quite different cultural horizons. The radiocarbon ages obtained from charcoal at different levels within the sediments do not provide reliable ages for the migratory stone artefacts within those sediments [118,126].

Two rock shelter sites in tropical northern Australia have yielded ages which are at least 15,000 years older than all previously dated sites in Australia. The claimed ages also run counter to genetic evidence which suggests that prehistoric humans did not reach Australia until about 45,000–50,000 years ago [4]. The very high dispersion rates obtained by single-grain luminescence dating at one of the sites are consistent with disturbance by mound-building termites, which are present at that site [5].

In higher latitudes, seasonal freezing and thawing of the soil can also cause vertical displacement of artefacts [127]. During the Last Glacial Maximum some 20,000 years ago, many areas beyond the margins of the ice sheets were subject to seasonal freezing and thawing, particularly in North America and Europe. The same processes also occur at high elevations. In such environments it is best to assume disturbance has been widespread. Dating archaeological sites in such regions should always be carried out by direct dating of the actual artefacts rather than relying on ages obtained from the sediments in which they occur.

## 4. Summary and Future Directions

Evidence from geomorphology has the great advantage that it is found in all continental environments. Examples of such evidence comes from rivers, lakes, desert dunes, caves and seacoasts. Changes in river behaviour linked to climatic changes in headwaters will influence the nature and distribution of archaeological sites downstream. The great tropical deserts of Africa, Asia and Australia have been intermittently wetter and more arid in the past. Lakes and rivers attracted human occupation during these wetter climatic intervals until aridity set in once more, and the rivers and lakes dried out. Alternating layers of wind-blown sand and river or lake sediments testify to these environmental fluctuations. On the Loess Plateau (Figure 1) of China, alternating layers of desert dust and fossil soil likewise indicate alternating drier and wetter climates, with wetter conditions proving attractive for prehistoric occupation. As the great continental ice caps waxed and waned, sea levels rose and fell, alternately providing new land for prehistoric settlement when sea levels were low and submerging archaeological sites when sea levels rose.

It is hard to predict future directions with any confidence because new techniques may revolutionise our future approach to archaeology, but some cautious suggestions can be made. One concerns the use of remote sensing data. Although there will never be an alternative to surveying on foot for archaeological sites and associated artefacts, fieldwork can be supplemented and, in some cases, even directed by the intelligent use of air photos, satellite imagery and other forms of remote sensing. Satellite images are not photographs but are compiled from recording different wavelengths of light reflected from the Earth's surface. By making use of different wavebands recorded in the digital imagery, it is possible to detect different types of soil, vegetation and rock, and even to detect buried former river channels based on subtle changes in soil temperature and moisture. Other forms of remote sensing include the use of radar. Shuttle radar has been used to locate buried river channels in the eastern Sahara, some of which contain artefacts ranging in age from Acheulian to Neolithic [128–130].

A far more integrated approach to excavation and analysis of archaeological sites and artefacts will be necessary in the future. The chronology and stratigraphic integrity of many previously excavated sites will probably require some form of re-appraisal, using a combination of micro-morphology, geochemistry and genetics. In particular, it will be important to show that the artefacts and fossil remains have been directly dated, and not just the sediments in which they occur. Far greater attention must be paid to the many, often quite subtle physical, chemical and biological processes that can cause disturbance within archaeological sites.

**Funding:** This research received no external funding.

**Institutional Review Board Statement:** Not applicable.

**Informed Consent Statement:** Not applicable.

**Data Availability Statement:** Not applicable.

**Acknowledgments:** My grateful thanks go to Frances Williams, who provided constructive and rigorous criticism of the text and converted my sketches into clear and elegant figures; to Patrick Williams, who helped improve my colour photographs; and to the two anonymous reviewers for their constructive suggestions. I dedicate this review to the memory of the late Liu Tungsheng, who pioneered loess studies in China and always made me welcome during my research visits to China.

**Conflicts of Interest:** The authors declare no conflict of interest.

## Glossary

*Geoarchaeology* is the use of earth science techniques in archaeology.
*Geomorphology* is the study of landforms and the processes responsible for their formation.
*Soil* is weathered rock and sediment in which plants grow.
The *Quaternary Period* is the most recent geological period, extending from 2.58 million years ago to present.
It includes the *Pleistocene Epoch* (2.58 Ma to 11.7 ka) and the *Holocene Epoch* (11.7 ka to present).

## Appendix A

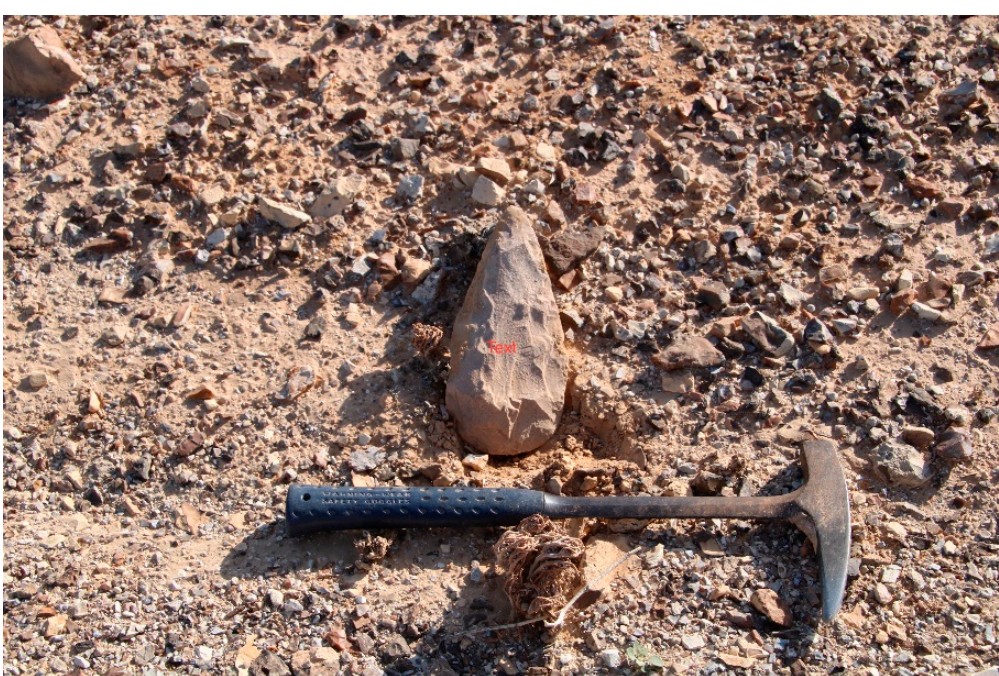

**Figure A1.** Lower Palaeolithic biface lying on a bare gravel surface, western Sahara.

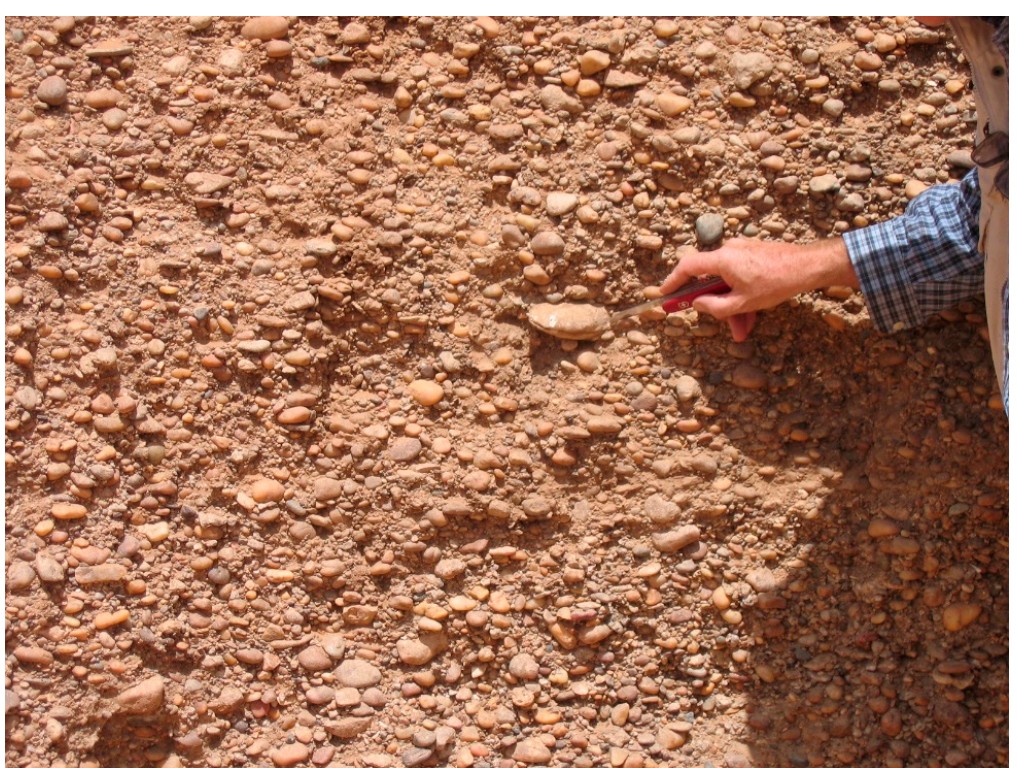

**Figure A2.** Alluvial pebbles and gravels laid down by the Nile in northern Sudan 20,000 years ago. The bivalve shells are a freshwater Nile mussel, most probably *Mutela* sp.

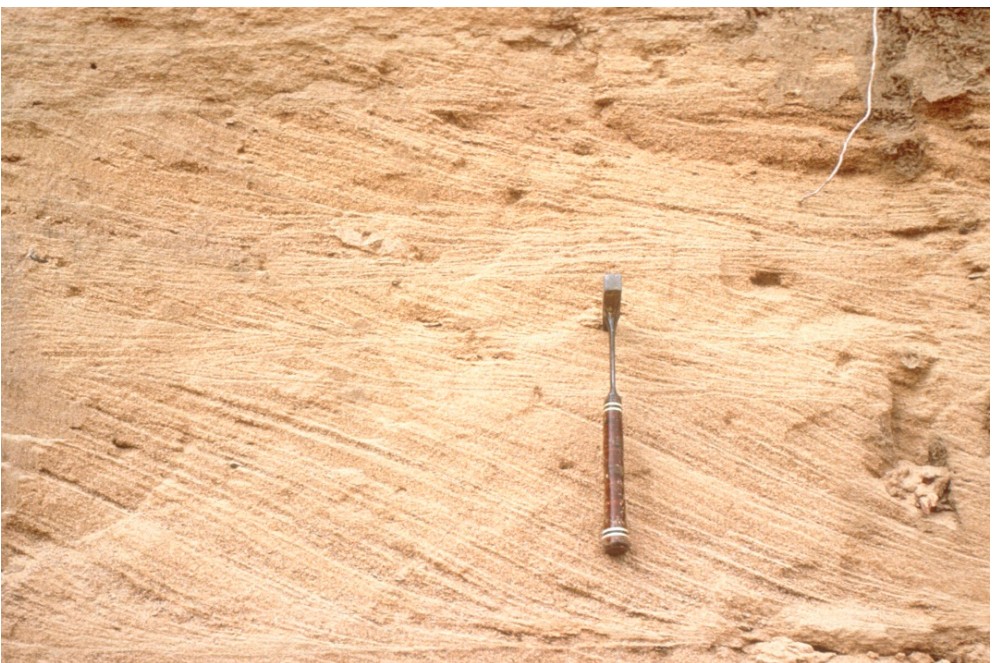

**Figure A3.** Cross-bedded sands of the Baghor Formation Lower Member, Middle Son valley, north-central India.

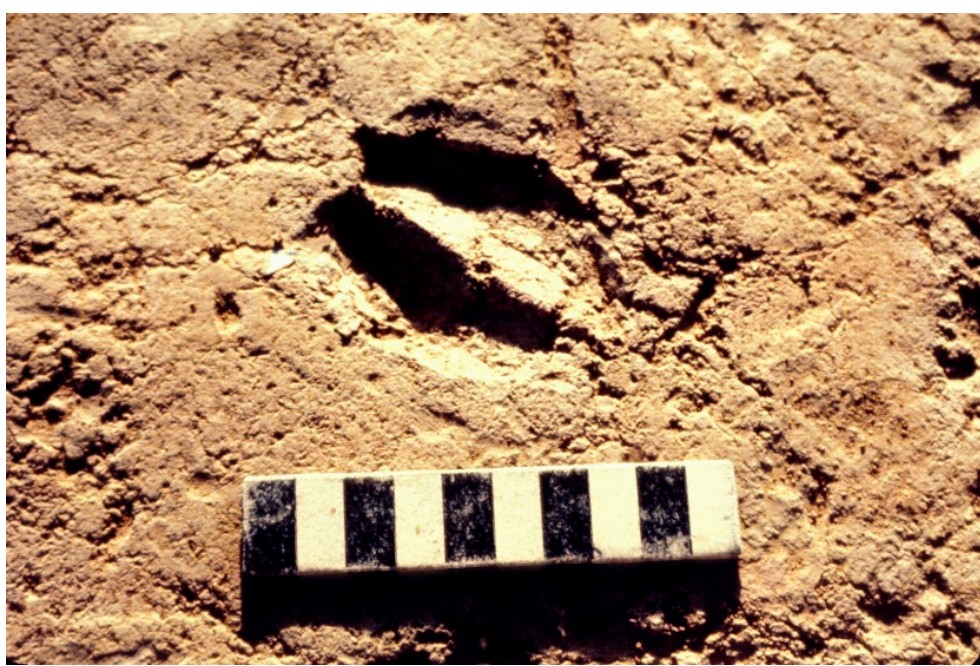

**Figure A4.** Prehistoric sambur deer tracks on the surface of a clay unit of the Baghor Formation Upper Member, Middle Son valley, north-central India, that was laid down 8000 years ago.

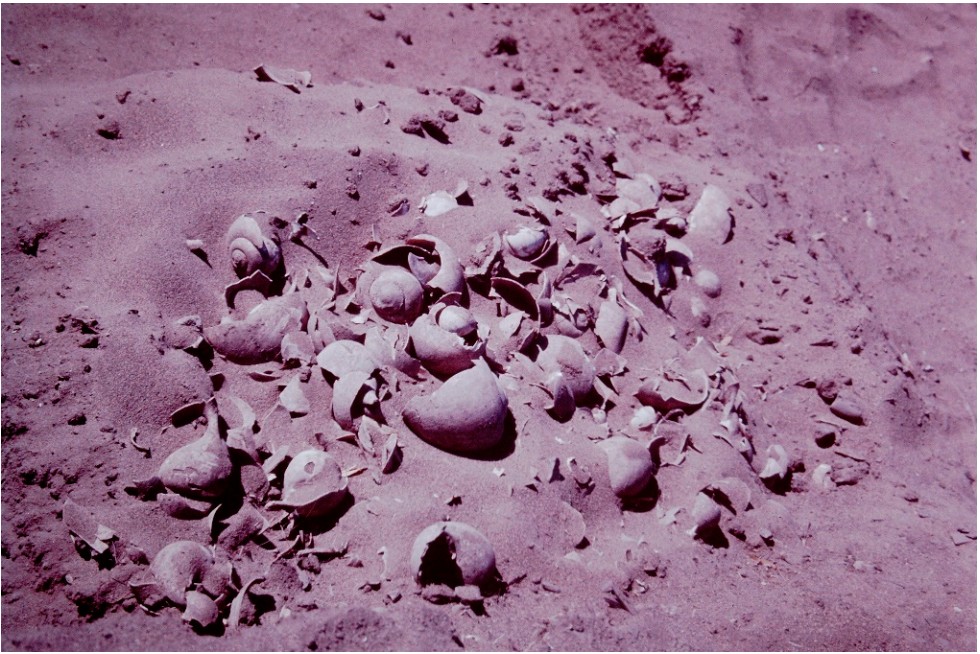

**Figure A5.** Mesolithic shell midden on the surface of a sand dune in the lower White Nile valley. The site was occupied at least seasonally about 8000 years ago. For details, see [83].

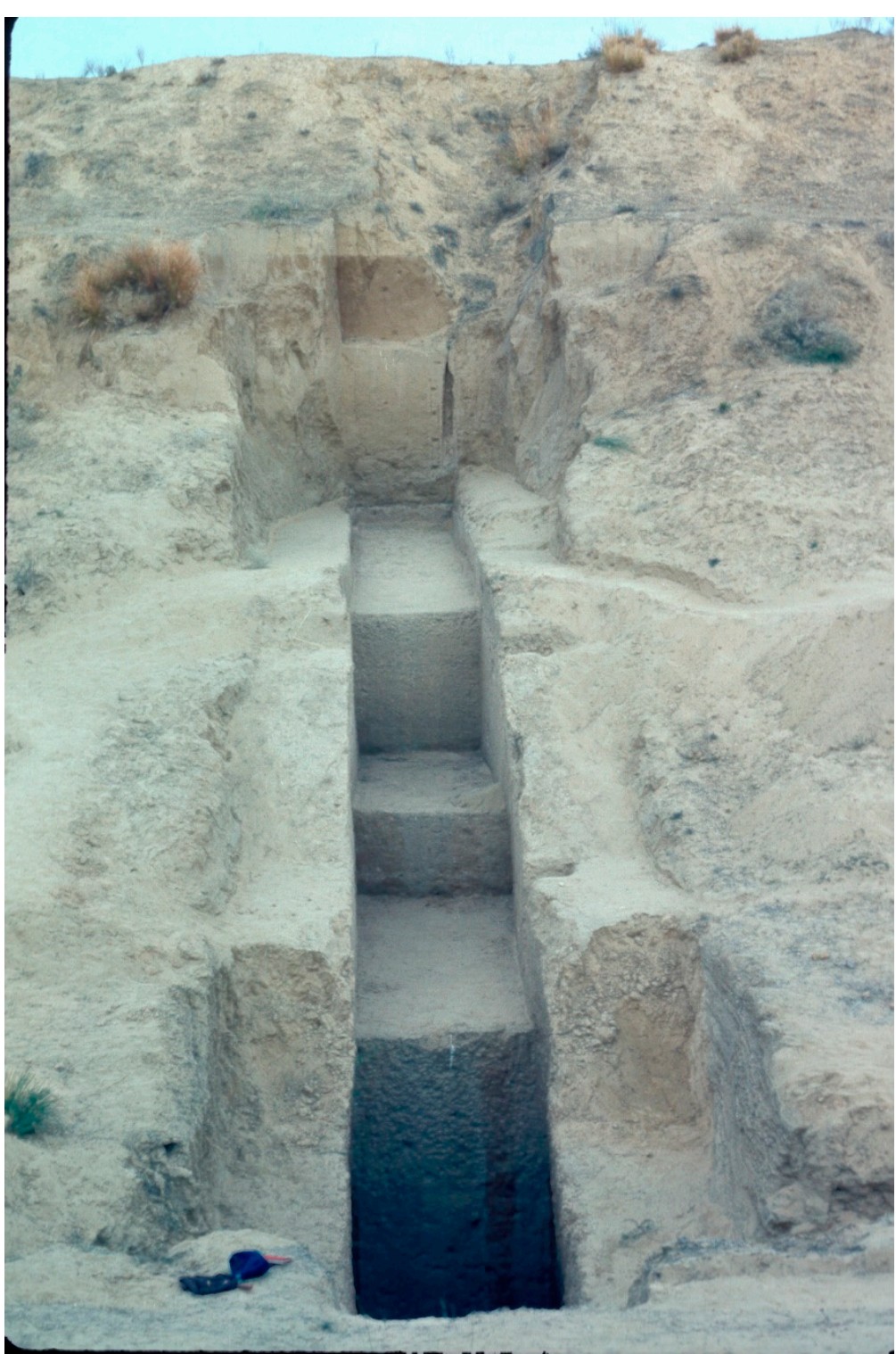

**Figure A6.** A trench 18.3 m deep was excavated through this dune site in the Thar Desert of northwest India. The dune consists of alternating buried soils cemented by calcium carbonate and wind-blown sand. The base of the trench has an age of about 200,000 years (see Figure 10).

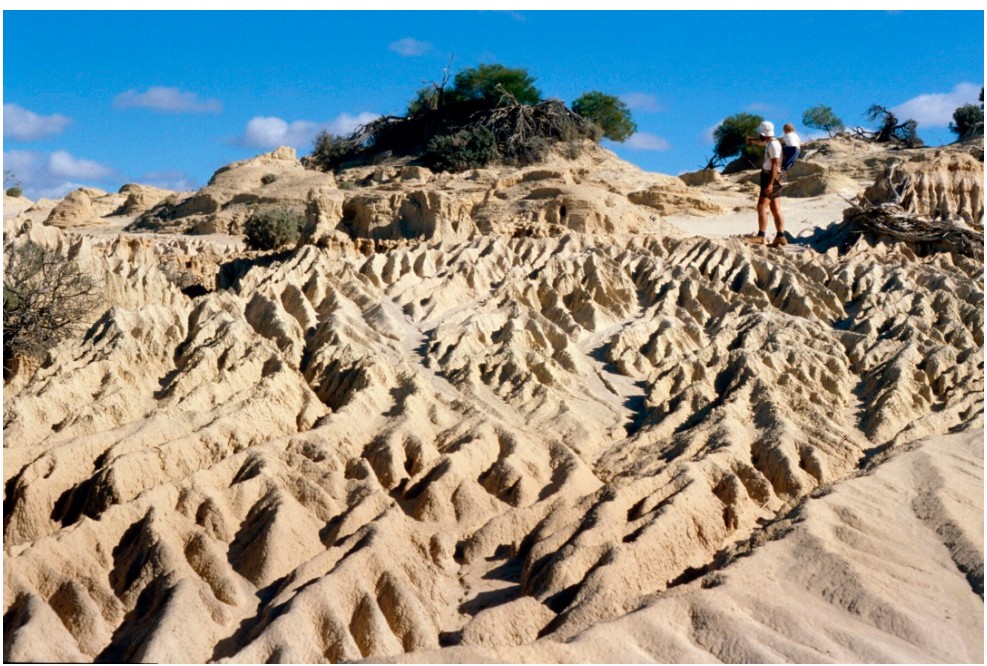

**Figure A7.** Eroded clay dune on the eastern margin of Pleistocene Lake Mungo.

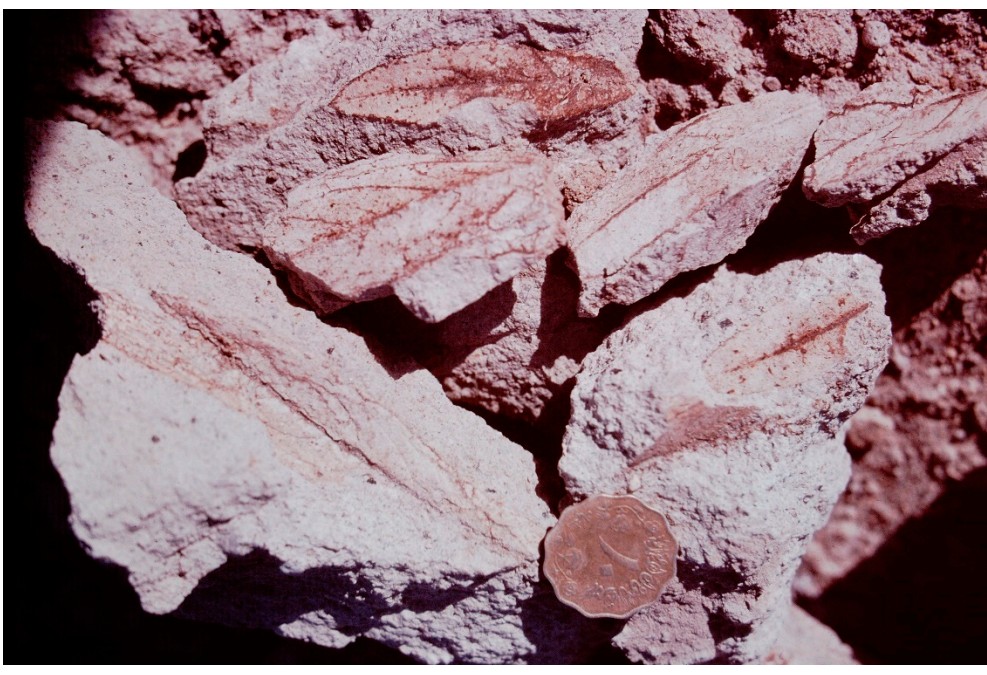

**Figure A8.** Oil palm leaf fossil preserved in volcanic ash south of the Jebel Marra volcano in arid western Sudan. The ash was erupted a million years ago during a wetter climatic phase.

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
