# Peer review of "When the Land Sings: Reconstructing Prehistoric Environments Using Evidence from Quaternary Geology and Geomorphology, with Examples Drawn from Fluvial Environments in the Nile and Son Valleys"

_quaternary, doi:10.3390/quat5030032_

Round 1

Reviewer 1 Report

Review

Journal: Quaternary

Title: When the land sings: Using evidence from geomorphology to reconstruct Quaternary prehistoric environments

Author: Martin Williams

Manuscript type: Review

Review date: 29th April 2022

The manuscript "When the land sings: Using evidence from geomorphology to reconstruct Quaternary prehistoric environments" was submitted to the scientific journal Quaternary and suggested in the Review category.

In addition to the title, abstract and glossary, the manuscript has 10 chapters and a list of references (42). It consists of 30 pages, ie 666 lines of text and 4 tables, 9 figures and 8 plates.

The main contribution of this paper is in the systematically presented geomorphological evidence related to environmental changes in prehistory. Emphasis is placed on river but also lake, desert, volcanic and coastal environments and geomorphological processes. The paper is based on numerous researches of the author of this article, but also other relevant literature.

Main disadvantages:

- For some data, paragraphs and claims, the cited literature is missing. This is why in some places it is more like a textbook than a review paper. I suggest expanding the bibliography a bit more since this is a review paper. Some examples of missing references are given in the detailed comments.

- All tables are missing a source citation - from which sources the tables are compiled

- Also, for some figures, the source of the images is not specified, and if it was drawn specifically for this article, then the source of the data should be specified.

Detailed comments:

L16 - should probably be Holocene / Neolithic

L18 - I propose one last concluding sentence on the importance of geomorphology within this topic

L156-161 - I am missing a reference

L241 - Table 4 - this table should be clarified in more detail in the text. Descriptions of several formations appear here, but their spatial context is not clear and the table is not sufficiently related to the text.

L300 – I am missing a reference

L332-339 – I am missing a reference

Figure 7 - a legend is needed (should it be indicated what brown is - clay?). Plan view and cross section do not match. In the plan the lunette appears asymmetrical, and in cross-section it is almost symmetrical.

Figure 8 – Plan view and cross section do not match. Eg - where is the alluvium in the plan? It would be better to unite the legend.

L414 - I think the word Discussion can be omitted from the title of Chapter 9. In fact, the chapter does not even discuss the results, they are discussed in the previous chapters, and here the issue of their disturbance is emphasized.

Author Response

I have acted on each of the comments below, adding new figures, new references and replacing certain figures with new figures. I have added more explanation to the figure captions as requested. 

I have sent the editor  the revised version of my review paper and trust that she will send both this and the revised figures and captions and references to you. All changes are shown in red. I attach the text, bibliography and captions for the Figures, Tables and Plates  separately from the actual Figures, Plates and Tables. 

Both reviewers requested location maps and many more references.

Figures

I have added four new location maps. Places names now shown on the new location maps are highlighted in yellow in the text.

Figure 1 is a world map showing places mentioned in the text that are not shown on the other location maps. The new Figure 5 covers the Nile basin and adjacent areas. The new Figure 7 shows the location of the Son and Belan rivers in India. The new Figure 9 shows the location of the Willandra lakes and Lake Mungo in Australia. This figure also replaces the previous Figure 7. The caption for Figure 11 (previously Figure 8) has been changed to indicate that 11a and 11b are not from the same locality. I have added explanatory captions for the previous Figures 1, 2, 3 and 4 (now respectively 2, 3, 4 and 6). Former Figure 5 is replaced with new Figure 8a and 8b. Former Figure 6 is replaced with new Figure 10. There are now 12 figures; previously there were 9, of which two have been replaced.

References

I have added 91 new references. The new references are at each place in the text requested by the two reviewers as well as at a few other places. Added to the previous 39 references, this brings the total to 130, which I consider more than sufficient to address the concerns of the two reviewers. All new references are shown in red.

Tables

I have added extra information to the captions of each Table.

Plates

I have added extra information to four of the Plate captions.

Omissions and additions

The title has been expanded as requested to read ‘When the land sings: Reconstructing prehistoric environments using evidence from Quaternary geology and geomorphology, with examples drawn from fluvial environments in the Nile and Son valleys.’

Other minor changes and additions are shown in red in the text.

I have expanded the acknowledgements to honour the memory of the late Professor Liu Tungsheng.

Reviewer 2 Report

Paper submitted by M. Williams is very interesting and should deserve a publication after minor/moderate changes. I appreciated the very detailed description of the geomorphological aspects/concepts applied to the different cases studies. The paper can be easily read by students or well experienced scientists.

Author is a very well experienced scientist in particular for the study of past landscapes/geomorphology of the Nile River. The authors is probably using the results/interpretations for the sites he investigated during his previous research activities. I could suggest to clearly separate/Evoke the sites studied by the author so the reader can referer to the previously published papers from the author.

  • Title should be changed and should focus more on the cases studies presented in the paper. I made a suggestion in the annotated pdf files I provide. Indeed, as it stands the title is very general and doesn't totally reflect the whole paper.
  • An important aspect of my criticism is addressed to the need to add more references, in particular for the introduction where it is highly expected to robustly document previous researches on the topic.
  • Another point concerns the lack of location maps of the mentioned sites. Two different scales must be adopted : a general (world scale with sites mentioned in Indonesia, Africa, etc.) one and a regional one for the well studied sites from the Nile and Son rivers.

For details, please check the pdf version where I reported my comments/suggestions of improvement.

Author Response

(The authors gave the same response as above.)
